# Is biological larviciding against malaria a starting point for integrated multi-disease control? Observations from a cluster randomized trial in rural Burkina Faso

**Peter Dambach**[1]*, **Till Bärnighausen**[1], **Anges Yadouleton**[2], **Martin Dambach**[3], **Issouf Traoré**[4], **Patricia Korir**[5], **Saidou Ouedraogo**[4], **Moustapha Nikiema**[4], **Rainer Sauerborn**[1], **Norbert Becker**[6], **Valérie R. Louis**[1]

1 Institute for Global Health, University Hospital Heidelberg, Heidelberg, Germany, 2 Centre de Recherche Entomologique de Cotonou (CREC), Cotonou, Benin, 3 Institute of Zoology, University of Cologne, Cologne, Germany, 4 Centre de Recherche en Santé de Nouna, Nouna, Burkina Faso, 5 Institute of Medical Microbiology, Immunology and Parasitology, University Hospital Bonn, Bonn, Germany, 6 German Mosquito Control Association (KABS), Speyer, Germany

* peter.dambach@uni-heidelberg.de

**Data Availability Statement:** The datasets supporting the conclusions of this article are publicly available in a public science repository

## Abstract

### Objectives

To evaluate the impact of anti-malaria biological larviciding with *Bacillus thuringiensis israelensis* on non-primary target mosquito species in a rural African setting.

### Methods

A total of 127 villages were distributed in three study arms, each with different larviciding options in public spaces: i) no treatment, ii) full or iii) guided intervention. Geographically close villages were grouped in clusters to avoid contamination between treated and untreated villages. Adult mosquitoes were captured in light traps inside and outside houses during the rainy seasons of a baseline and an intervention year. After enumeration, a negative binomial regression was used to determine the reductions achieved in the different mosquito species through larviciding.

### Results

Malaria larviciding interventions showed only limited or no impact against *Culex* mosquitoes; by contrast, reductions of up to 34% were achieved against *Aedes* when all detected breeding sites were treated. *Culex* mosquitoes were captured in high abundance in semi-urban settings while more *Aedes* were found in rural villages.

### Conclusions

Future malaria larviciding programs should consider expanding onto the breeding habitats of other disease vectors, such as *Aedes* and *Culex* and evaluate their potential impact.

under the following doi: https://doi.org/10.5281/zenodo.4732822.

**Funding:** This study was funded by the Manfred Lautenschläger foundation, Wiesloch, Germany. The funder did not have any role in the design implementation and the analysis of the study.

**Competing interests:** The authors have declared that no competing interests exist.

**Abbreviations:** *Bti*, *Bacillus thuringiensis israelensis*; *Bs*, *Bacillus sphaericus*; LTC, Light Trap Captures.

Since the major cost components of such interventions are labor and transport, other disease vectors could be targeted at little additional cost.

## Introduction

Larval source management, involving the elimination, alteration, and treatment of breeding grounds of disease transmitting mosquitoes, has been practiced for centuries. During the 1950s insecticides such as DDT and Paris Green had become a promising tool for global malaria eradication, but they were later abandoned because of their disastrous effect on the environment. Today´s malaria vector control targets predominantly the adult stages of mosquitoes through bed nets and indoor residual spraying. The larviciding approach involves biological substances that are not harmful for the environment. Routine implementation is carried out predominantly in high income countries, but several large-scale trials have been carried out in urban and rural Africa during the last two years and have proved technically feasible and demonstrated an impact on malaria vector populations. Evidence of their impact on actual malaria transmission is currently backed by only a few studies [1, 2]. Promising results were achieved with the bacterial toxins *Bacillus thuringiensis israelensis* (*Bti*) and *Bacillus sphaericus (Bs)* that act selectively against mosquitoes and are environmentally sound.

Although the primary-target during anti-malarial larviciding interventions are mosquitoes from the genus *Anopheles*, there is an impact on other mosquito genera that inhabit the same breeding sites. Within the study region in North-Western Burkina Faso the typical malaria mosquito breeding sites consist of water holes, brickworks, small ponds, wet rice fields and large, flooded areas. During the peak rainy season puddles can persist for up to several weeks and allow for mosquito breeding. Apart from a wide variety of *Anopheles* species, those breeding sites are equally attractive for oviposition to female *Culex* mosquitoes and there is evidence that, despite observed inter species predation [3], both *Anopheles spp*. and *Culex spp*. are more likely to coexist in the same breeding sites than would be expected by chance alone [4, 5]. While there is a major overlap in breeding site preference between *Anopheles* and *Culex* mosquitoes, various species of *Culex* were found to be generally more successful breeding in heavily polluted water bodies. Within the study region, those heavily polluted sites prevail in the semi-urban town of Nouna, mainly as septic tanks and dirty puddles, while they are almost absent in the rural villages. *Aedes* mosquitoes on the other hand normally prefer other types of breeding sites that are not primary targets of larviciding interventions against malaria. Typical breeding sites of the regionally common vector *Aedes aegypti* include drinking water containers, clay jars, tin cups, car tires and other small objects that can harbor rainwater.

Although the highly abundant *Culex* and *Aedes* mosquitoes are not capable of transmitting human malaria, they do have increasing public health relevance in Africa through the transmission of several arboviral and parasitic infections. Culex mosquitoes are known to transmit West-Nile fever (*Culex pipiens*), Sindbis-virus (*C. pipiens*, *C. univittatus*) and parasitic nematodes such as *Wucheria bancrofti*, a cause of lymphatic filariasis. Several species from the genus *Aedes* are known to transmit the dengue, Chikungunya, and yellow fever viruses. Both *Culex* and *Aedes* can transmit the Zika virus. To date there are numerous mosquito-borne arboviruses transmitted by *Culex* and *Aedes* mosquitoes that are indigenous to Africa, and several of them are likely to receive greater geographical distribution and significance for health with increasing population growth, travel, and deforestation [6]. *Mansonia* do have public health relevance as some species are capable of transmitting lymphatic filariasis [7].

Given the high vector competence and capacity for transmitting several emerging diseases of at least some of these genera, it is important to know how much they are affected by malaria vector control interventions. However, to date, there is only very limited knowledge about cross benefits of malaria larviciding programs on other disease vectors [8], mostly because species other than *Anopheles* have not been considered during impact evaluation. In this study we evaluate the extent to which non-malaria mosquito populations are impacted by *Bti* based larviciding against malaria vectors in the public space, in and around 127 rural villages and a semi-urban town in North-Western Burkina Faso.

## Methods

### Study area

The study area consisted of all 127 rural villages and the semi-urban town that form part of the extended health district of the Kossi region in Northwestern Burkina Faso, close to the Mali border. It stretches over a total surface of about 4,770 km$^2$ and contains some 156,000 inhabitants. The area is characterized by two distinct seasons, a dry season that extends from November to April, and a rainy season between late June and October. The study area is heterogeneous in its ecology. While the Northern parts towards the Sahel often feature sandy soils with high infiltration rates and lower numbers of environmental mosquito breeding sites, the terrain is different in the South, where there are more stagnant water bodies and wet rice growing areas. The Eastern border of the district is characterized by wetlands around the Sourou Valley.

### Study design

The study was designed as a cluster randomized trial, administering different larviciding options to mosquito breeding sites [9]. Reporting followed the CONSORT guidelines for randomized trials where applicable. Three larviciding options (i: untreated control, ii: treatment of all breeding sites and iii: risk map based larvicide application) were performed within a total of 9 village clusters (Fig 1). The risk map-based application of larvicides used data on *Anopheles* larval densities and related it to such water parameters as turbidity, presence of algae, and vegetation cover, which were identifiable via remote sensing on satellite images. In this study arm, only half of breeding sites received *Bti*-based larviciding, those assessed as most productive by the model, while the less productive half was omitted during treatment. The approach is described in more detail elsewhere [9, 10]. Villages were clustered to avoid spill-over effects caused by the flight range of mosquitoes [11, 12]. Three clusters consistently represented areas that were similar in surface water availability, soil type, vegetation and other geographical factors (ecozone). Larviciding options were randomly assigned to the predefined clusters, ensuring that each larviciding option was represented in each geographical ecozone (Fig 1). The study lasted three years, consisting of a baseline year without intervention (2013) and two intervention years (2014 + 2015). Here we present results from the baseline and the first intervention year, in which the abundance of non-anophelines was determined in addition to *Anopheles* mosquitoes. Larviciding with *Bti* VectoBac® WG, AM65-52 strain (Valent BioSciences Corporation, IL, USA) was performed during and up to six weeks after the rainy season in the public space of the villages and in a 500 m buffer zone around villages but not in private compounds. Spraying in all villages took place every ten days and was followed by a quality control test through dipping for living larvae the day after. VectoBac® was diluted in pond water filtered through cotton cloth and brought out onto the water surface using inox steel knapsack sprayers (Mesto®, Freiberg, Germany). Prior to the intervention the optimum dosages for field application were identified and larviciding was then carried out at 0.35kg/ha (equaling 0.35mg/l).

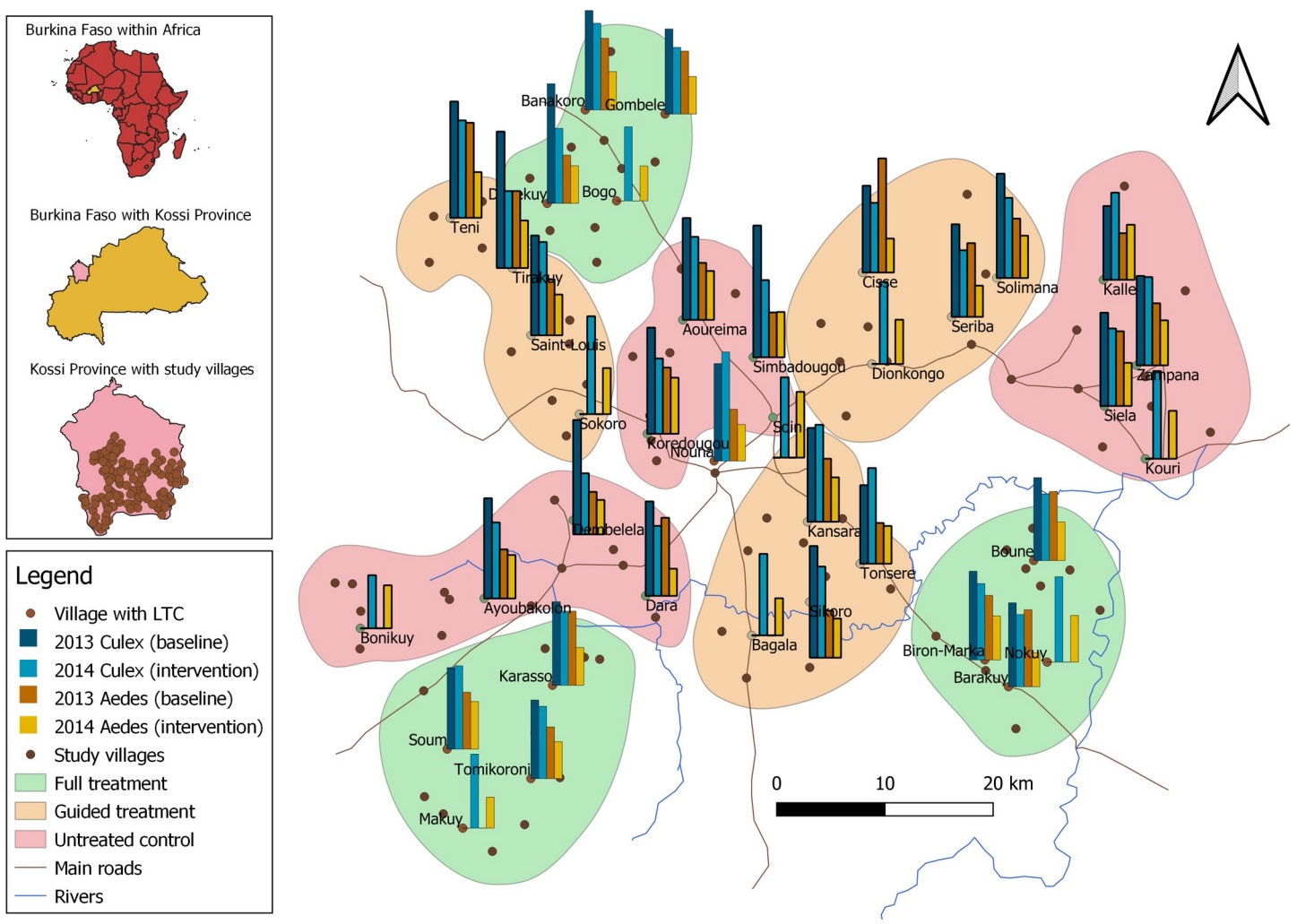

**Fig 1. Mosquito genera distribution in baseline and intervention year.** Study villages are shown with dark brown dots; villages with light trap captures (LTC) are marked with light brown dots. Bars show the average numbers of female *Culex* (blue color range) and female *Aedes* (orange color range) captured per trap per night both indoors and outdoors in September and October 2013 and 2014. Colors of the cluster areas indicate treatment option (green = full treatment, orange = guided treatment, red = untreated control). In 2014, 9 additional villages were added to the LTC mosquito collections.

## Adult mosquito monitoring

The primary outcome used to assess larviciding efficacy was the abundance of different mosquito species. For the collection of adult mosquitoes, Center for Disease Control light traps (Model 512, John W. Hock Company, Gainesville, Florida) were used. Indoor and outdoor captures of *Aedes* and *Culex* mosquitoes were performed in 27 villages in 2013, and in 36 villages in 2014; additionally, the seven town quarters of Nouna were included. Light trap captures were performed every two weeks, following a rotating system with two independent fieldwork teams, covering 4 villages per night, resulting in a total of at least 10 sample rounds per village per rainy season.

In each study village, three households (compounds) were chosen for their central position in the village and in agreement with the household head. Light traps were installed approximately 100 to 150 meters from each other to detect possible local differences in vector abundance between different places within one village. Each a light trap was positioned about one

meter above the ground. The traps inside houses were installed near the sleeping places equipped with untreated bed nets, the traps outside were placed beside the house within the common courtyard, where people sat in the evenings. Mosquitoes were collected between 18:00 and 06:00 hours to fully cover the peak biting period.

## Mosquito species identification

For the *Aedes* and *Culex* mosquito genera, a total of 122 and 150 specimens, respectively, from several villages were characterized at the level of sibling species. The species of the *Aedes* mosquitoes was determined by microscopy, using morphological criteria. *Culex* mosquitoes were identified to sibling species level using PCR, following the protocols of Kasai and Smith-Fonseca [13, 14]. The primers used were: ACE pip2: 5′GGTGGAAACGCATGATACCAG 3′, ACE quin: 5′CCTTCTTGAATGGCTGTGGCA 3′, F1457: 5′GAGGAGATGTGGAATCCCAA 3′, B1246s: 5′TGGAGCCTCCTCTTCACGG 3′. A reaction mixture for each PCR tube consisted of 6.2 μl of water; 10 μl of AmpliTaq GoldTM 360 Master Mix; 0.4 μl of ACE pip2 primer (0.2 μM); 0.8 μl of ACE quin primer (0.4 μM); 0.8 μl of primer F1457 (0.4 μM) and 0.8 μl of primer B1256s (0.4 μM).

## Statistical analysis

Statistical analysis was performed using Stata/IC 14.2 for Windows (StataCorp LLC, 4905 Lakeway Drive, College Station, TX 77845, USA). The count of female mosquitoes collected per night per trap was used as the outcome variable. This nonnegative count variable showed overdispersion and was thus modelled using a negative binomial regression (Stata function "nbreg"), which corresponds to a generalization of a Poisson distribution that accommodates a variation greater than that of a true Poisson. A parsimonious regression model was selected because it sufficiently captured data variation. The model regressed the mosquito count per genera against the *Bti* treatment options (untreated, guided and full) and the temporal variable of sample round to account for seasonal variations (corresponding to approximately two "batches" per month); a random effect was included at village level to take spatial clustering in consideration.

## Ethics approval and consent to participate

This study was approved by the ethics committees of the University of Heidelberg in Germany, the national ethics board of Burkina Faso in Ouagadougou and the local ethics committee at the research site in Nouna. Aggregated collective verbal informed consent for the spraying activities was collected for each village through the traditional village chiefs, with at least one additional person from the village and two responsible persons from the research team being present. The population was invited at a central place in the village and the project, its goals and the activities involved were explained in local language. Following this, public discussions were held with the opportunity to ask questions or express concern. During the intervention there was additional community sensitization and information, performed through the local radio station. The study was registered under the trial id PACTR201611001721299 on the Pan African Clinical Trials Registry (https://pactr.samrc.ac.za).

## Consent for publication

There are no case presentations that require disclosure of respondents' confidential data/information in this study.

## Results

### Mosquito species distribution before and during larviciding

A total of 36,148 female mosquitoes were captured using light traps between September and December 2013 and between June and November 2014. During the four months of sampling in the pre-intervention year, 12,073 female mosquitoes were caught, of which 5,842 (48%) were *Culex* spp. (Linnaeus), 3,677 (30%) *Anopheles* spp. (Meigen), 2,317 (19%) *Aedes* spp. (Meigen) and 237 (2%) *Mansonia* spp. (Blanchard). For the two genera that have possible public health relevance for the transmission of vector borne diseases other than malaria, *Culex* and *Aedes*, PCR and morphological analysis showed that the analyzed samples consisted predominantly of only one sibling species. For *Culex*, 97% of specimens included in the sample were *Culex pipiens quinquefasciatus*, for *Aedes*, all specimens were *Aedes aegypti aegypti*. During the six month of mosquito collections in the intervention year, 24,075 female mosquitoes were captured; the share of *Culex* mosquitoes on the total catch increased to 55% (13,205), while the abundance of *Anopheles* decreased to 23% (5,345). The share of *Aedes* remained almost unchanged with 22% (5,357) of the total catch while that of *Mansonia* decreased to 0.7% (168). Fig 1 illustrates the geographical variation in *Culex quinquefasciatus* and *Aedes aegypti aegypti* mosquito numbers among villages and in the semi-urban town of Nouna during the annual period of high mosquito abundance.

### Outdoor and indoor captures

Fig 2 shows mosquito abundance by genus and by place of capture (indoors or outdoors) in the different treatment areas. *Culex* and *Aedes* mosquitoes were predominantly captured indoors (54% and 57%, respectively). The difference in number between indoor and outdoor capture was statistically significant only for *Culex* mosquitoes (p = 0.026) and for *Aedes* mosquitoes (p = 0.071) the counts displayed a large variability. However, the much fewer *Mansonia* mosquitoes were largely captured outdoors (75%, p = 0.007). In the semi-urban area of Nouna, *Culex* mosquitoes were highly abundant despite being in an area of full treatment.

### Reduction of non-*anophelines* through *Bti*- spraying

Fig 3 shows that environmental larviciding with *Bti* only had no significant effect on the abundance of *Culex* mosquitoes (1.06, 95% CI: 0.96–1.16 and 0.96, 95% CI: 0.88–1.06, for guided and full treatment, respectively) while the abundance of *Aedes* mosquitoes was significantly reduced by 34% with the full treatment (0.66, 95% CI: 0.57–0,76) but not with the guided treatment (0.94, 95% CI: 0.85–1.05).

### Mosquito abundance over time

The abundance of *Culex* and *Aedes* mosquitoes followed the course of the rainy season. *Culex* populations began to rapidly grow with a lagged onset of about two weeks after the first rains in July, reaching a maximum in August, and then slowly declined until the end of the capture period in late November (Fig 4). All rural areas, whether under full, guided or no treatment, showed similar abundance patterns of *Culex* during the rainy season, underlining the absence of a significant impact of larviciding on this genus at all the times observed. Despite larviciding interventions having been in place with the first rains in beginning July, *Culex* catches peaked at almost 16 mosquitoes per night per trap in August. For *Aedes* the picture was the reverse, the larviciding interventions showed greatest impact during the month of August and the highest reductions achieved were registered in the semi-urban setting of Nouna town, where

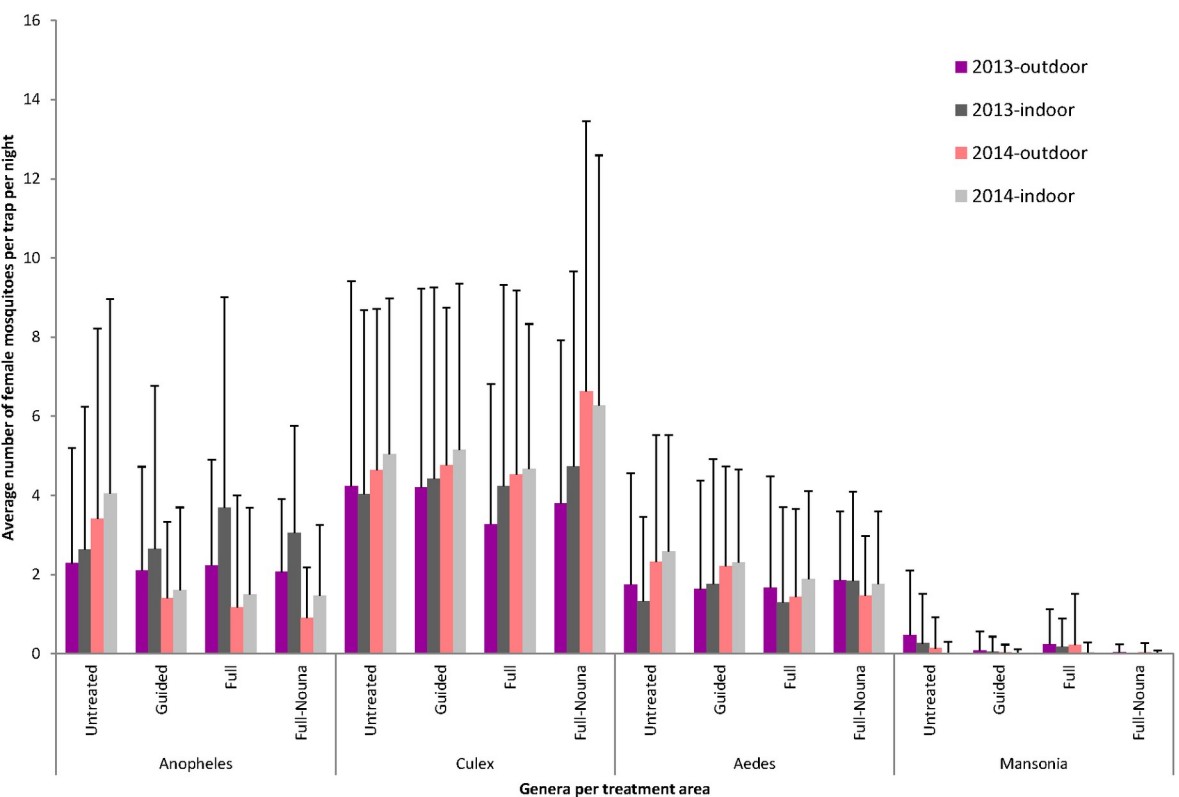

**Fig 2. Average number of female mosquitoes per trap per night, per genus and treatment area for indoor and outdoor LTC capture over the collection period in 2013 and 2014.** The error bars indicate the standard deviation within each group.

catches were as low as 0.5 mosquitoes per night per trap. However, towards the later months September and October, the number of *Aedes* did rise again.

## Discussion

While the abundance of adult *Anopheles* spp. was suppressed by up to 70%, the same larviciding intervention did not show a significant reduction in the overall impact on the abundance of *Culex quinquefasciatus* mosquitoes. Reductions of *Aedes aegypti aegypti* were low in the guided treatment arm but attained 34% in the full treatment arm. When looking at the achieved reductions for each of the two prevailing mosquito species stratified by semi-urban or rural environment, one observes a more diverse picture. While the larviciding intervention focused only on typical malaria mosquito breeding sites within the public space, its collateral effects on *Culex quinquefasciatus* and *Aedes aegypti aegypti* were inverse. In the full treatment study arm in the semi-urban town of Nouna, the interventions showed comparably high reduction of *Aedes aegypti aegypti*, while there was no impact on *Culex quinquefasciatus*. Inversely, in the rural study villages, the impact on *Culex quinquefasciatus* was higher, compared to only little alteration in *Aedes aegypti aegypti*. The virtually non-existing impact of larviciding activities on *Culex quinquefasciatus* mosquitoes in the semi-urban arm of our study matches findings from Dar es Salaam in Tanzania, where only little effect of larviciding on adult *Culex* was achieved, while the primary target *Anopheles* was strongly impacted. *Culex quinquefasciatus* are known to breed even in heavily polluted water and we observed the same within the survey region, where heavily polluted oviposition sites such as septic tanks, oily

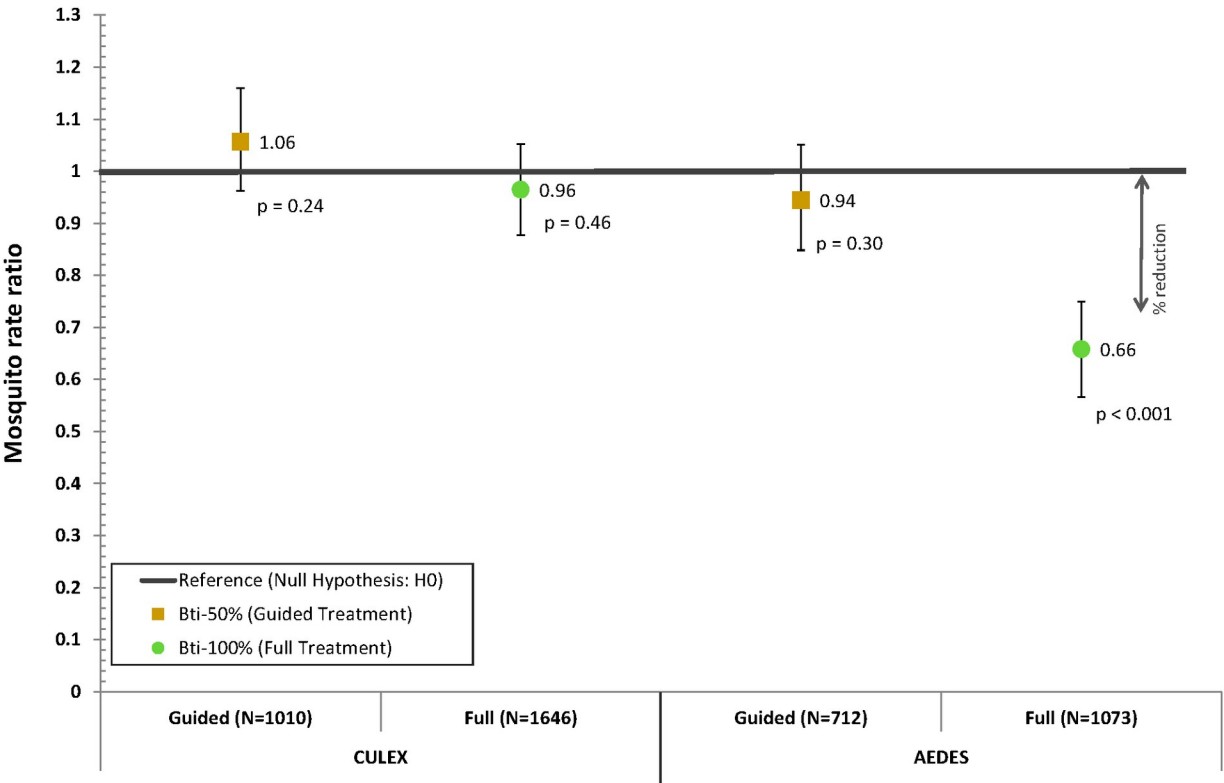

**Fig 3. Point estimates of the regression model for the intervention year compared to the baseline year indicating the reduction in the counts of female *Culex* and *Aedes* mosquitoes achieved through different larviciding options with *Bti*, i.e. guided (*Bti*-50%) or full treatment (*Bti*-100%) of *Anopheles* larval habitats treated in rural villages, excluding the town of Nouna.** The reference line represents the rate ratio value under the null hypothesis: i.e. the count of female mosquitoes in the control areas receiving no *Bti* treatment are not significantly different from the counts in areas receiving a guided or full *Bti* treatment.

puddles and open surface toilets were exclusively populated with *Culex* larvae. Within the semi-urban setting, those breeding sites were mostly situated within private courtyards, which were not targeted by spraying activities. An explanation for better *Culex quinquefasciatus* reduction in rural villages might be that they there share the less polluted, larger habitats with *Anopheles* mosquitoes, while habitats exclusive to *Culex quinquefasciatus*, such as pit latrines and polluted puddles, are much less prevalent in the rural environment.

Not only did the differences in recorded mosquito reductions depend on whether the setting was rural or semi-urban, but also on whether the captures with light traps were performed at indoor or outdoor posts. Although for *Anopheles* the reductions at indoor capture post were twice as high compared to those from outdoor posts, the effect on non-*Anopheles* mosquitoes showed the reverse picture, with twice the reductions achieved at outdoor posts [15, 16]. It is difficult to conclude what led to the higher *Culex quinquefasciatus* and *Aedes aegypti aegypti* reductions at outdoor posts. A previous study in the region that used human landing catches found *Anopheles gambiae* s.l. to be the predominant species. with a share of more than 90 percent of the total *Anopheles* catch. The species (or sub-species) did not seem to be a relevant factor in determining the degree of attraction to either outdoor or indoor LTC posts; geographical location, however, was found to be a factor [17]. Despite possible differences in species composition and LTC trap preference in different villages, these factors are insufficient to explain differences in achieved reductions between outdoor and indoor LTC posts. Reductions achieved through targeting mosquito larvae would be expected to appear uniformly,

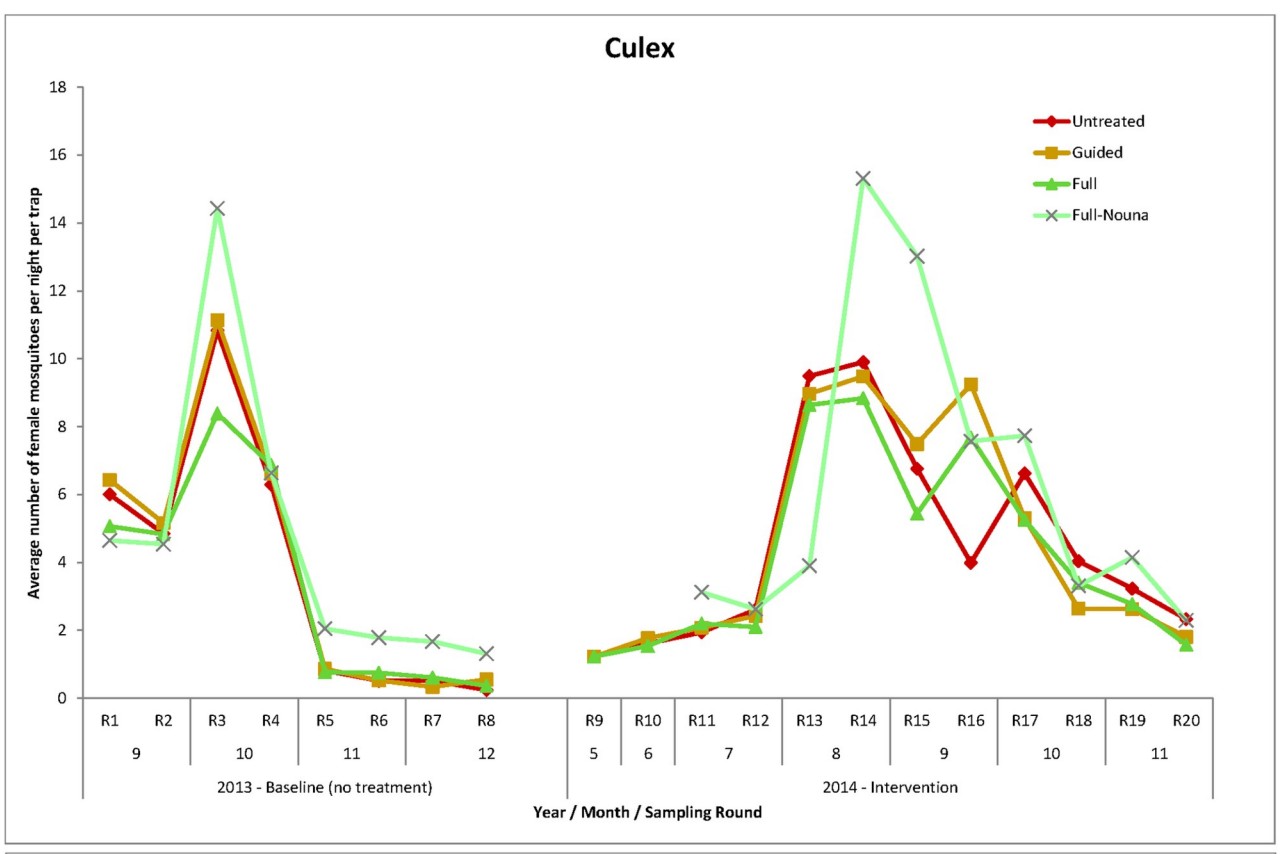

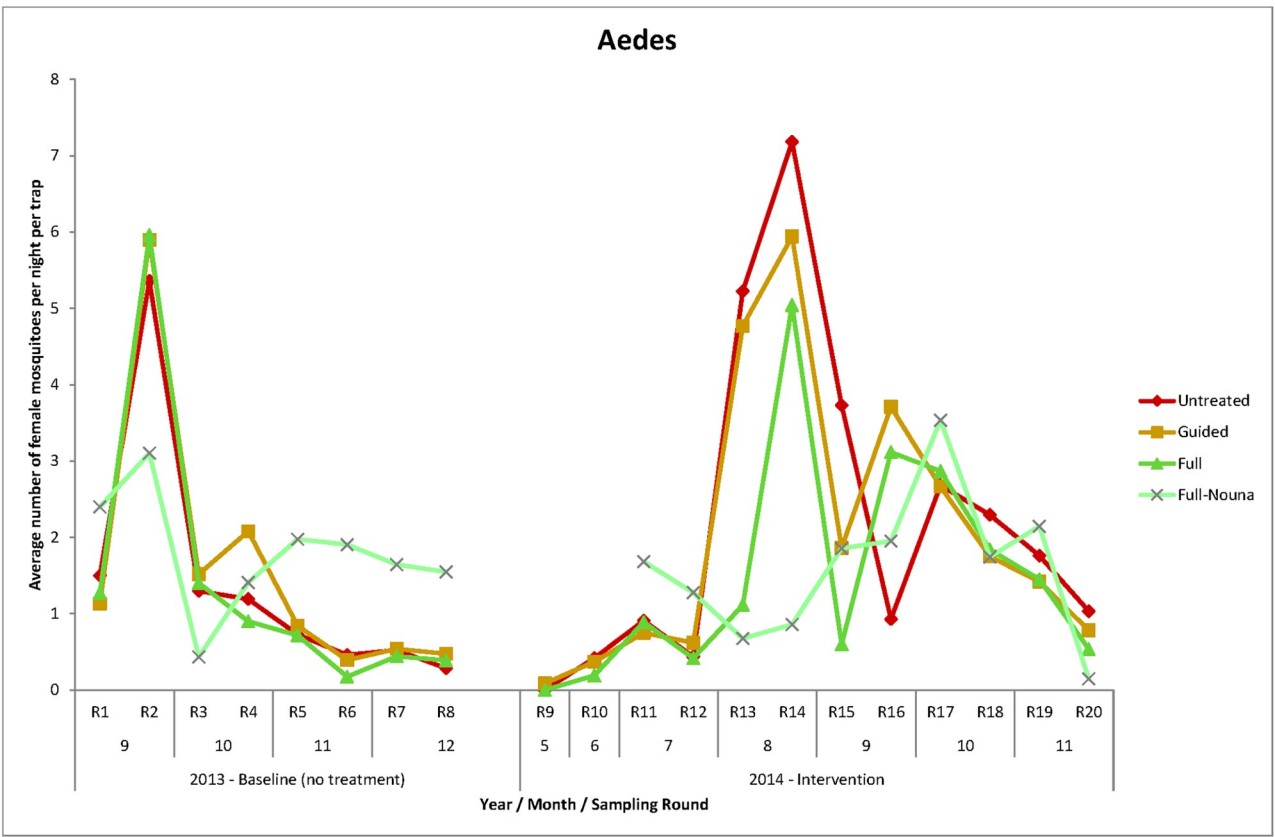

**Fig 4.** Timeline of mosquito abundance (average number of female mosquitoes per trap per night) in the different *Bti* treatment areas, A) Culex mosquitoes (*Culex pipiens quinquefasciatus*) B) Aedes mosquitoes (*Aedes aegypti aegypti*). The error bars indicate the standard deviation within each group.

unless different mosquito species are attracted differently to outdoor and indoor LTC posts or larviciding interventions affect different species differently. The use of light traps seemed to be more effective in capturing indoor resting *Culex*, *Aedes* and *Anopheles* mosquitoes, while the traps positioned outdoors showed generally lower mosquito numbers. This contrasts with another study in the same area that used human landing catches and found the abundance of all three genera between indoor and outdoor to be roughly the same [18].

Theoretically, difference in larviciding success between different mosquito genera might be ascribed not only to different habitat types but also to their individual susceptibility to *Bti*. However, reports on the effectiveness of *Bti* on the larvae of different genera within the *Culicidae* family differ. While some studies reported higher susceptibility of *Culex* towards *Bti*, others found that *Anopheles* and *Aedes* required lower lethal *Bti* concentrations. Pollution and eutrophication of breeding sites is known to influence the effectiveness of *Bti*. If breeding in sites with increased pollution, they might have been less affected by spray activities.

### Strengths and limitations

This study benefits from the large spatial and temporal extent of larviciding activities in 127 rural villages and a semi-urban town and the high amount and collection frequency of entomological data, which is extensive compared to many other studies [9]. To our knowledge this study is the first to systematically evaluate the impact of larviciding against malaria vectors on other disease transmitting mosquito genera. There are also limitations to this study. Mosquito collections in 2013 started later than initially planned and data is available from September on only, resulting in a relatively short overlap period of three months with the mosquito sampling of the following intervention year. For the determination of species level for *Aedes* and *Culex* mosquitoes, a limited sample size of 122 and 150 specimens, respectively, was used to determine the species distribution within the region. The mosquito samples for each genus were composed of specimens from different villages and different collection periods. Despite the diversity of origin in time and place, the identified species showed little diversity with 100% of *Aedes* and 97% of *Culex* belonging to only one species. This high homogeneity suggests that an increased sample size may have yielded similar results. This observation is supported by various occurrence records reported by Harbach [19]. Treatment arms were randomized at the level of village clusters. Although this does not follow the standard approach for a randomized control trial of medical studies, it was the best possible approach in a geographical and environmental context. Mosquitoes not only bite in the immediate vicinity of their breeding grounds but are able to travel some distance. Those distances can differ by genus, species, weather, and type of environment [20–24]. As the primary interest of the underlying study was malaria vector control, we applied larviciding over a larger area to avoid infiltration of mosquitoes from untreated areas, since Anopheles were reported to travel up to several kilometers in rural areas [11, 12]. For this reason, villages in which the same larviciding approach was applied were clustered geographically.

### Importance for vector control programs

The findings presented here have several implications for the implementation of larviciding programs for mosquito control. Mosquito species other than *Anopheles* are not relevant in the context of malaria but they do play a role in nuisance and in other vector borne diseases such

as lymphatic filariasis, yellow fever, dengue, and Zika. Although malaria control programs that are limited to performing larviciding in the public space, such as the one presented here, can provide major reductions in *Anopheles* mosquito abundance, they often lack the ability to sufficiently reduce other disease transmitting mosquitoes, such as various species of *Culex* and *Aedes*. These develop in breeding sites that are typically found in private compounds [25]. Whereas our study targeting *Anopheles* mosquitoes in public places showed a reduction on *Aedes* but not on *Culex*, it raises the question about the possibility of extending larviciding to private compounds. While we expect only limited additional relief in *Anopheles gambiae* abundance when extending spraying activities to private compounds, we could anticipate a strong impact on the numbers of the predominantly abundant *Culex pipiens quinquefasciatus* and *Aedes aegypti aegypti* mosquitoes, especially in the semi-urban area. However, in contrast to such other vector control interventions as insecticide treated nets and curtains, which have been shown to work against different vector borne diseases [26], for larviciding there is a lack of robust epidemiological studies evaluatinge the impact on diseases other than malaria [27].

In addition to the actual impact on other vector borne diseases, the treatment against *Culex* and *Aedes* mosquitoes could contribute to an increased acceptability and support of vector control programs. This is because most community members are not able to distinguish between mosquitoes that transmit diseases and those that do not, but they identify and value a general reduction in mosquito abundance and nuisance. Regardless of whether an epidemiological impact on diseases other than malaria is intended, or the perceived program success is maximized through the reduction of nuisance mosquitoes, the additional impact would need to be put into an economically meaningful relation with the additionally required resources.

## Conclusion

In the wake of the introduction of such vector borne diseases as dengue to Africa, it could become important not to limit vector control efforts to *Anopheles* but to extend them to disease transmitting species of *Culex* and *Aedes* as well. Since the major cost components of larviciding based vector control programs are infrastructural and personnel expenditures, it could be beneficial to bundle efforts for controlling malaria, dengue and other mosquito-borne diseases into an integrated program where necessary. Future studies need to investigate what impact larviciding interventions of private compounds could achieve, considering that they contain a multitude of habitats for various disease-relevant *Aedes* and *Culex* species.

## Acknowledgments

We are deeply thankful to the communities for their support and willingness to participate in this research. We are also grateful to the field and laboratory staff at the research facility in Nouna for their valuable work and commitment to make the project successful and evolving.

## Author Contributions

**Conceptualization:** Peter Dambach, Till Bärnighausen, Martin Dambach, Issouf Traoré, Rainer Sauerborn, Norbert Becker.

**Data curation:** Peter Dambach, Issouf Traoré, Valérie R. Louis.

**Formal analysis:** Peter Dambach, Till Bärnighausen, Valérie R. Louis.

**Investigation:** Peter Dambach, Issouf Traoré.

**Methodology:** Martin Dambach, Valérie R. Louis.

**Project administration:** Peter Dambach, Saidou Ouedraogo.

**Software:** Till Bärnighausen, Valérie R. Louis.

**Supervision:** Peter Dambach, Martin Dambach, Issouf Traoré.

**Validation:** Peter Dambach, Moustapha Nikiema.

**Visualization:** Peter Dambach.

**Writing – original draft:** Peter Dambach, Till Bärnighausen, Anges Yadouleton, Martin Dambach, Issouf Traoré, Patricia Korir, Saidou Ouedraogo, Rainer Sauerborn, Valérie R. Louis.

**Writing – review & editing:** Peter Dambach, Anges Yadouleton, Martin Dambach.

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
