## [Decision Letter · Decision Letter 0]

19 Mar 2021

PONE-D-20-34035

Is Biological Larviciding against Malaria a Starting Point for Integrated Multi-Disease Control? – Observations from a Cluster Randomized Trial in rural Burkina Faso.

PLOS ONE

Dear Dr. Dambach,

Thank you for submitting your manuscript to PLOS ONE. After careful consideration, we feel that it has merit but does not fully meet PLOS ONE’s publication criteria as it currently stands. Therefore, we invite you to submit a revised version of the manuscript that addresses the points raised during the review process.

We look forward to receiving your revised manuscript.

Kind regards,

Nicholas C. Manoukis

Academic Editor

PLOS ONE

Additional Editor Comments:

Thank you for your submission- this paper represents substantial field monitoring and treatment application effort, and I believe it provides useful information to programs that include larviciding aiming to control mosquito vectors in W Africa. Please address comments by reviewers.

I find the writing to be understandable but awkward in places- please check through again. In addition, I did not see statistical models presented, though there are p-values in the results section- Please add tables with negative binomial regression results- also provide more information on this model (I am confused as to the mention of a "random effect"- I infer this a mixed model? I feel like more details are needed on statistical approach). On a similar theme, Figures 2 and 4: I think these should include error bars or indication of CI unless I badly misunderstand something.

I am looking forward to seeing a revision.

Journal Requirements:

5. Thank you for stating the following in the Funding Section of your manuscript:

[This study was funded by the Manfred Lautenschläger foundation, Wiesloch, Germany. The funder did not have any role in the design implementation and the analysis of the study.]

 [The author(s) received no specific funding for this work.]

6. We note that Figure 1 in your submission contain map images which may be copyrighted. All PLOS content is published under the Creative Commons Attribution License (CC BY 4.0), which means that the manuscript, images, and Supporting Information files will be freely available online, and any third party is permitted to access, download, copy, distribute, and use these materials in any way, even commercially, with proper attribution. For these reasons, we cannot publish previously copyrighted maps or satellite images created using proprietary data, such as Google software (Google Maps, Street View, and Earth). For more information, see our copyright guidelines: http://journals.plos.org/plosone/s/licenses-and-copyright.

You may seek permission from the original copyright holder of Figure 1 to publish the content specifically under the CC BY 4.0 license. 

If you are unable to obtain permission from the original copyright holder to publish these figures under the CC BY 4.0 license or if the copyright holder’s requirements are incompatible with the CC BY 4.0 license, please either i) remove the figure or ii) supply a replacement figure that complies with the CC BY 4.0 license. Please check copyright information on all replacement figures and update the figure caption with source information. If applicable, please specify in the figure caption text when a figure is similar but not identical to the original image and is therefore for illustrative purposes only.

Reviewers' comments:

Reviewer's Responses to Questions

**Comments to the Author**

1. Is the manuscript technically sound, and do the data support the conclusions?

Reviewer #1: Yes

Reviewer #2: Partly

2. Has the statistical analysis been performed appropriately and rigorously? 

Reviewer #1: Yes

Reviewer #2: I Don't Know

3. Have the authors made all data underlying the findings in their manuscript fully available?

Reviewer #1: Yes

Reviewer #2: Yes

4. Is the manuscript presented in an intelligible fashion and written in standard English?

Reviewer #1: Yes

Reviewer #2: Yes

5. Review Comments to the Author

Reviewer #1: Overall, this is a very straightforward manuscript about the effects of larviciding on the abundance of 2 genera of mosquitoes (the larviciding is originally targeted at a third kind of mosquito in many areas, including this study area). Interestingly, the number of mosquitoes increased from the control year to the treatment year at control sites, and this was reflected in the Culex data as well (further indicating the lack of effect on this genus) - according to Figure 1. I am trying to see how Figure 1 and Figure 2 line up - so it might be easier on the reader to make the colors match between the figures (and even Figure 4).

There are a few copyedit suggestions, check for correct use of hyphens in certain locations (e.g., lines 52, 54, 85) and adding commas in other locations (e.g., lines 56, 83) - not all mentioned here.

I recommend with minor revision - doing some strong copyediting and possibly making the colors of the graphs more uniform so that the reader can more easily connect the results across the graphs.

Reviewer #2: The manuscript requires major revisions particularly in the English composition, cohesion and inclusion of relevant details and references (see comments below). The acceptance of this paper is conditional. The study was done in 2013 and 2014 and only now the authors have submitted this manuscript. Although, WHO still recommends the use of Vectobac WG (AM65-52 strain) as a mosquito larvicide, the authors must show the gap of pertinent studies in the recent past to justify the novelty of their findings.

The manuscript is partly fine, however, it is poorly written and inconclusive. They concluded that:

“Future larviciding programs should be evaluated for including the treatment of Aedes and Culex breeding habitats. Since the major cost components of such programs are labor and transport, other disease vectors could be targeted at little additional cost”.

1) However, their data showed that Baccillus thurigiensis israelensis (Bti) larviciding had limited or no impact against Culex mosquitoes. Their major finding was Bti larviciding reduced up to 34% of Aedes mosquitoes in breeding sites in public spaces.

2) Based on their indoor and outdoor captures of adult mosquitoes, they found that Anopheles were reduced at indoor captures twice as high compared to those from outdoors (lines 224-226). But they were not certain what factors that led to higher reduction of Culex quinquefasciatus and Aedes aegypti at outdoors (lines 224-228).

The above major findings might be the reason that the title poses a question implying that the results are not conclusive and their underlying purpose of this submission apparently is to report their observations of their first year of intervention year. Hence, they “concluded” with a recommendation to further evaluate future larviciding programs for including the treatment of Aedes and Culex breeding sites with the use of Bti larvicide.

Note that the scope of this manuscript included only the baseline and the first intervention year. The paper might be more conclusive if the results of the second intervention year would be included to be more certain on their conclusion.

I suggest that a statistician’s opinion can further elucidate the judgment of their statistical analyses. My comments include:

1. Lines 169-171: Please describe clearly and explicitly the interpretations of the statistical analyses performed. What were the Culex mosquitoes significant of? How about Aedes and Mansonia?

2. Lines 257-263: The sample size bias of 122 Aedes and 150 Culex mosquitoes was justified by stating Harbach (2012). I am not certain whether their justification is statistically acceptable. What I am certain is in their research design, the power analysis of sample size was not predetermined prior to actual field work, hence, they had to retrospectively justify what they got during the actual field work.

Add additional relevant references that can augment the use of Bti in setting the background of your study. Were there related studies in the recent past that used Bti against malaria mosquitoes in other African countries and in Asia? How novel is your work particularly that your study was done in 2013 and 2014 yet? Lines 46-49: Reference 1 is published in 2007 and is not current anymore, inconsistent of the sentence in these lines.

Lines 52, 56, 61-62, 75, 210-214, 219, etc.: Improve grammar on the agreement between subject and verb, and sentence construction.

Methods:

1. Please include a GIS-map of the study site. Indicate the months of the two seasons in the study sites.

2. Line 98, 239, 245-246, 252: References please. Lines 267-268: Please cite relevant references indicating the average flight distances of each mosquito species included in the study. Avoid generalizing them.

3. Lines 99-101: Please describe briefly but concisely the third larviciding choice (risk map-based larvicide). What do you mean by the most productive breeding sites? Please cite references.

4. Line 142: Please indicate the primers and PCR-reaction mixture used in identifying the sibling species of Culex even though you cited Kasai (2008) and Smith-Fonseca (2004).

6. PLOS authors have the option to publish the peer review history of their article (what does this mean?). If published, this will include your full peer review and any attached files.

Reviewer #1: No

Reviewer #2: No

---

## [Author Response · Author response to Decision Letter 0]

2 May 2021

Response to Reviewers

PONE-D-20-34035

Is Biological Larviciding against Malaria a Starting Point for Integrated Multi-Disease Control? – Observations from a Cluster Randomized Trial in rural Burkina Faso.

PLOS ONE

Dear Dr. Manoukis, dear reviewers,

Many thanks for your valuable input through comments and proposed amendments to improve this manuscript. Below please find a point to point response to reviewer comments. 

We made the data underlying the analysis freely available through a public science repository, with its own doi. We changed the data availability statement accordingly. 

We removed the Funding statement from the Declarations. If possible, we would like to have the following statement to be included: “This study was funded by the Manfred Lautenschläger foundation, Wiesloch, Germany. The funder did not have any role in the design implementation and the analysis of the study”.

 We hope that this revised version of the manuscript is open for consideration to be published in PLOS ONE.

Sincerely,

Peter Dambach

Additional Editor Comments:

Thank you for your submission- this paper represents substantial field monitoring and treatment application effort, and I believe it provides useful information to programs that include larviciding aiming to control mosquito vectors in W Africa. Please address comments by reviewers.

I find the writing to be understandable but awkward in places- please check through again. In addition, I did not see statistical models presented, though there are p-values in the results section- Please add tables with negative binomial regression results- also provide more information on this model (I am confused as to the mention of a "random effect"- I infer this a mixed model? I feel like more details are needed on statistical approach). On a similar theme, Figures 2 and 4: I think these should include error bars or indication of CI unless I badly misunderstand something.

The “Statistical analysis” section describing the model has been greatly expanded to better explain and clarify the procedure use to model the data. We feel that adding the negative binomial regression table would be too cumbersome within this manuscript. As requested, we added error bars in figure 2; for figure 4 we would prefer not to display the error bars because it clusters the figure too much. 

Journal Requirements:

Our manuscript and the file names now follow the PLOS ONE style requirements.

We have added the information on the type of consent and the number of people present during informed consent. It now reads as follows: “Aggregated collective verbal informed consent for the spraying activities was collected for each village through the traditional village chiefs, with at least one additional person from the village and two responsible persons from the research team being present”. The amended ethics statement will be equally added in the submission form. 

We made the underlying database accessible to the reader. It is stored in a public science repository under the following doi: https://doi.org/10.5281/zenodo.4732822. The section “Availability of data and material” now reads as follows: “The datasets supporting the conclusions of this article are publicly available in a public science repository under the following doi: https://doi.org/10.5281/zenodo.4732822”.

We moved the ethics statement (updated as per reviewer point 2) to the end of the Methods section.

5. Thank you for stating the following in the Funding Section of your manuscript:

[This study was funded by the Manfred Lautenschläger foundation, Wiesloch, Germany. The funder did not have any role in the design implementation and the analysis of the study.]

 [The author(s) received no specific funding for this work.]

We removed the Funding statement from the Declarations. If possible, we would like to have the following statement to be included: “This study was funded by the Manfred Lautenschläger foundation, Wiesloch, Germany. The funder did not have any role in the design implementation and the analysis of the study”.

6. We note that Figure 1 in your submission contain map images which may be copyrighted. All PLOS content is published under the Creative Commons Attribution License (CC BY 4.0), which means that the manuscript, images, and Supporting Information files will be freely available online, and any third party is permitted to access, download, copy, distribute, and use these materials in any way, even commercially, with proper attribution. For these reasons, we cannot publish previously copyrighted maps or satellite images created using proprietary data, such as Google software (Google Maps, Street View, and Earth). For more information, see our copyright guidelines: http://journals.plos.org/plosone/s/licenses-and-copyright.

We created a new map for figure 1, using our own vector-data. We removed the underlying base map. Copyrights are completely with the first-author and allows it to be used under the CC BY 4.0 license.

Reviewers' comments:

Reviewer #1: 

Overall, this is a very straightforward manuscript about the effects of larviciding on the abundance of 2 genera of mosquitoes (the larviciding is originally targeted at a third kind of mosquito in many areas, including this study area). Interestingly, the number of mosquitoes increased from the control year to the treatment year at control sites, and this was reflected in the Culex data as well (further indicating the lack of effect on this genus) - according to Figure 1. I am trying to see how Figure 1 and Figure 2 line up - so it might be easier on the reader to make the colors match between the figures (and even Figure 4).

There are a few copyedit suggestions, check for correct use of hyphens in certain locations (e.g., lines 52, 54, 85) and adding commas in other locations (e.g., lines 56, 83) - not all mentioned here.

I recommend with minor revision - doing some strong copyediting and possibly making the colors of the graphs more uniform so that the reader can more easily connect the results across the graphs.

The manuscript was fully edited by a native English speaker (see tracked changes) and we are confident that the text reads much better.

All figures were revised to harmonize the color schemes and make them uniform among figures when possible. (Note that the color choices are also compatible for black and white printing or color-blind readers). Fig 1 and Fig 2, however, are presenting the results in different ways, it is there’re not possible to use the same colors. Thus, to avoid confusion we chose different colors for Fig and Fig 2 to indicated clearly the values shown cannot be directly compared. Nonetheless, for both figures we chose colors that were darker in 2013 and lighter in 2014.

Reviewer #2: 

The manuscript requires major revisions particularly in the English composition, cohesion and inclusion of relevant details and references (see comments below). 

The manuscript was fully edited by a native English speaker (see tracked changes) and we are confident that the text reads much better.

The acceptance of this paper is conditional. 

The study was done in 2013 and 2014 and only now the authors have submitted this manuscript. Although, WHO still recommends the use of Vectobac WG (AM65-52 strain) as a mosquito larvicide, the authors must show the gap of pertinent studies in the recent past to justify the novelty of their findings.

The reviewer raises an important point, namely the age of the data, underlying our analysis and the novelty of the results. To our knowledge there is not much literature available that systematically looks at the effects of anti-malaria larviciding on other disease vector mosquitoes. We agree that this needs to be pointed out and added the following sentence and a reference to the last section of the Introduction: “However, to date, there is only very limited knowledge about cross benefits of malaria larviciding programs on other disease vectors (8), mostly because species other than Anopheles have not been looked at during impact evaluation”.

The manuscript is partly fine, however, it is poorly written and inconclusive. They concluded that:

“Future larviciding programs should be evaluated for including the treatment of Aedes and Culex breeding habitats. Since the major cost components of such programs are labor and transport, other disease vectors could be targeted at little additional cost”.

1) However, their data showed that Baccillus thurigiensis israelensis (Bti) larviciding had limited or no impact against Culex mosquitoes. Their major finding was Bti larviciding reduced up to 34% of Aedes mosquitoes in breeding sites in public spaces.

2) Based on their indoor and outdoor captures of adult mosquitoes, they found that Anopheles were reduced at indoor captures twice as high compared to those from outdoors (lines 224-226). But they were not certain what factors that led to higher reduction of Culex quinquefasciatus and Aedes aegypti at outdoors (lines 224-228).

The above major findings might be the reason that the title poses a question implying that the results are not conclusive and their underlying purpose of this submission apparently is to report their observations of their first year of intervention year. Hence, they “concluded” with a recommendation to further evaluate future larviciding programs for including the treatment of Aedes and Culex breeding sites with the use of Bti larvicide.

The reviewer raises an important point and we revised the discussion so that it better reflects the results of the paper and better explains the logic behind the recommendation. The text was modified both in the abstract and in the discussion.

Abstract: “Future malaria larviciding programs should consider expanding onto the breeding habitats of other disease vectors, such as Aedes and Culex and evaluate their potential impact. Since the major cost components of such interventions are labor and transport, other disease vectors could be targeted at little additional cost”

Discussion “Whereas our study targeting Anopheles mosquitoes in public places showed a reduction on Aedes but not on Culex, it raises the question about the possibility of extending larviciding to private compounds”.

Note that the scope of this manuscript included only the baseline and the first intervention year. The paper might be more conclusive if the results of the second intervention year would be included to be more certain on their conclusion.

We agree with the reviewer that this would further strengthen the results. However, since the primary focus of the project were malaria transmitting mosquitoes (so were the allotted funds and laboratory capacities), we determined species other than Anopheles only in the baseline (2013) and the first intervention year (2014). We specified this in the second sentence of the section “Adult mosquito monitoring” within the results. It now reads as follows, with added part in italics: “Indoor and outdoor captures of Aedes and Culex mosquitoes were performed in 27 villages in 2013, and in 36 villages in 2014; additionally, the seven town quarters of Nouna were included”.

I suggest that a statistician’s opinion can further elucidate the judgment of their statistical analyses. My comments include:

1. Lines 169-171: Please describe clearly and explicitly the interpretations of the statistical analyses performed. What were the Culex mosquitoes significant of? How about Aedes and Mansonia?

We agree with the reviewer; L169-171 were poorly written and the text has been edited so that the results are expressed in a clearer way. The text now reads as follow: “Figure 2 shows mosquito abundance by genus and by place of capture (indoors or outdoors) in the different treatment areas. Culex and Aedes mosquitoes were predominantly captured indoors (54% and 57%, respectively). The difference in number between indoor and outdoor capture was statistically significant only for Culex mosquitoes (p=0.026) and values for Aedes mosquitoes (p= 0.071) displayed a large variability.”

2. Lines 257-263: The sample size bias of 122 Aedes and 150 Culex mosquitoes was justified by stating Harbach (2012). I am not certain whether their justification is statistically acceptable. What I am certain is in their research design, the power analysis of sample size was not predetermined prior to actual field work, hence, they had to retrospectively justify what they got during the actual field work.

We agree with the reviewer and it is correct that the sample size was not predetermined prior to actual field work with regard to Aedes and Culex species. The reason is that the study was designed for Anopheles control, as explained. The results presented here report interesting additional information regarding non-target mosquito species. Although not ideal from a statistical point of view, this procedure is valid and checks retrospectively that the results are nonetheless statistically significant. Regarding the specific limitation in lines 257-263, we reformulated the text to make the point about the limited number of mosquitoes analyzed clearer. 

“For the determination of species level for Aedes and Culex mosquitoes, a limited sample size of 122 and 150 specimens, respectively, was used to determine the species distribution within the region. The mosquito samples for each genus were composed of specimens from different villages and different collection periods. Despite the diversity of origin in time and place, the identified species showed little diversity with 100% of Aedes and 97% of Culex belonging to only one species. This high homogeneity suggests that an increased sample size may have yielded similar results. This observation is supported by various occurrence records reported by Harbach”

Add additional relevant references that can augment the use of Bti in setting the background of your study. Were there related studies in the recent past that used Bti against malaria mosquitoes in other African countries and in Asia? How novel is your work particularly that your study was done in 2013 and 2014 yet? Lines 46-49: Reference 1 is published in 2007 and is not current anymore, inconsistent of the sentence in these lines.

We updated the references on the impact of larviciding trials on malaria transmission by two recent systematic literature reviews from 2018 and 2019 and gave a brief overview of the currently proven efficacy of larviciding on malaria. We also deleted the sentence on limitations in floodplains and its corresponding reference from 2007 and corrected the previous and following for grammar and cohesion. The section from (previously) line 45 onwards, now reads as follows, with added parts in italics: “However, several large-scale trials that have shown technical feasibility and an impact on malaria vector populations were carried out in urban and rural Africa during the last couple of years. Evidence for an impact of larviciding on malaria transmission as primary outcome is currently only backed by a few studies (1,2)”.

Lines 52, 56, 61-62, 75, 210-214, 219, etc.: Improve grammar on the agreement between subject and verb, and sentence construction.

Methods:

1. Please include a GIS-map of the study site. Indicate the months of the two seasons in the study sites.

We agree with the reviewer that a better overview of where the study region would be useful to the reader. For not having too many figures in the publication, we decided to update figure 1 and include a detailed overview map (composed of three inset maps), which includes the position of Burkina Faso within Africa, the position of the Kossi Province within Burkina Faso, and a minimap of the Kossi Province with all study villages. 

We added information on the two rainy seasons to the first paragraph of the methods section, it now reads as follows: “The area is characterized by two distinct seasons, a dry season that extends from November to April, and a rainy season between late June and October”.

2. Line 98, 239, 245-246, 252: References please. Lines 267-268: Please cite relevant references indicating the average flight distances of each mosquito species included in the study. Avoid generalizing them.

We added relevant references at the above-mentioned positions (not marked as track changes though, to avoid problems with reference manager software and the track changes function). For lines 267-268, we specified that flight ranges can differ by genus and various other factors and made clear that our choice of buffer radius was based on the flight range of Anopheles mosquitoes. We added references for both. It now reads as follows, with the added parts in italics: “Mosquitoes do not only bite in the immediate vicinity of their breeding grounds but are able to travel some distance. Those distances can differ by genus, species, weather, and type of environment (18–22). As the primary interest of the underlying study was malaria vector control, we applied larviciding over a larger area to avoid infiltration of mosquitoes from untreated areas, since Anopheles were reported to travel up to several kilometers in rural areas (9,10)”.

3. Lines 99-101: Please describe briefly but concisely the third larviciding choice (risk map-based larvicide). What do you mean by the most productive breeding sites? Please cite references.

We reckon that it might be confusing to already specify (“to only the most productive breeding sites”) when listing the three different larviciding choices. Since a more in-depth description of the generation of risk maps is given in lines 101-106. We removed the “to only the most productive breeding sites”. Additionally, we added in line 106 “The approach is described in more detail elsewhere”, followed by two references that describes the modelling approach of risk-map development and the detailed meaning of the 50% most productive cut off threshold.

4. Line 142: Please indicate the primers and PCR-reaction mixture used in identifying the sibling species of Culex even though you cited Kasai (2008) and Smith-Fonseca (2004).

We added the required information to “Mosquito species identification” within the Methods, it now reads as follows: “The primers used were: ACE pip2: 5’GGTGGAAACGCATGATACCAG 3’, ACE quin: 5’CCTTCTTGAATGGCTGTGGCA 3’, F1457: 5’GAGGAGATGTGGAATCCCAA 3’, B1246s: 5’TGGAGCCTCCTCTTCACGG 3’. A reaction mixture for each PCR tube consisted of 6.2 μl of water; 10 μl of AmpliTaq GoldTM 360 Master Mix; 0.4 μl of ACE pip2 primer (0.2 μM); 0.8 μl of ACE quin primer (0.4 μM); 0.8 μl of primer F1457 (0.4 μM) and 0.8 μl of primer B1256s (0.4 μM)”.

---

## [Decision Letter · Decision Letter 1]

9 Jun 2021

Is Biological Larviciding against Malaria a Starting Point for Integrated Multi-Disease Control? – Observations from a Cluster Randomized Trial in rural Burkina Faso.

PONE-D-20-34035R1

Dear Dr. Dambach,

We’re pleased to inform you that your manuscript has been judged scientifically suitable for publication and will be formally accepted for publication once it meets all outstanding technical requirements.

Kind regards,

Nicholas C. Manoukis

Academic Editor

PLOS ONE

Additional Editor Comments (optional):

Congratulations on a very nice revision. The paper is, in my view, acceptable for publication in PLOS ONE. The funding statement should be fine in the Acknowledgments, unless editorial staff have other information.

Reviewers' comments:

Reviewer's Responses to Questions

**Comments to the Author**

1. If the authors have adequately addressed your comments raised in a previous round of review and you feel that this manuscript is now acceptable for publication, you may indicate that here to bypass the “Comments to the Author” section, enter your conflict of interest statement in the “Confidential to Editor” section, and submit your "Accept" recommendation.

Reviewer #1: All comments have been addressed

2. Is the manuscript technically sound, and do the data support the conclusions?

Reviewer #1: Yes

3. Has the statistical analysis been performed appropriately and rigorously? 

Reviewer #1: Yes

4. Have the authors made all data underlying the findings in their manuscript fully available?

Reviewer #1: Yes

5. Is the manuscript presented in an intelligible fashion and written in standard English?

Reviewer #1: Yes

6. Review Comments to the Author

Reviewer #1: My previous comments regarding readability / English language and grammar have been well-addressed. The figures have also been improved, with a more consistent color scheme (if still not perfectly harmonized across all figures). The statistical questions raised by the other reviewer appear to have been addressed as well, and additional references/citations included.

7. PLOS authors have the option to publish the peer review history of their article (what does this mean?). If published, this will include your full peer review and any attached files.

Reviewer #1: No

---

## [Editor Report · Acceptance letter]

11 Jun 2021

PONE-D-20-34035R1 

Is Biological Larviciding against Malaria a Starting Point for Integrated Multi-Disease Control? – Observations from a Cluster Randomized Trial in rural Burkina Faso. 

Dear Dr. Dambach:

I'm pleased to inform you that your manuscript has been deemed suitable for publication in PLOS ONE. Congratulations! Your manuscript is now with our production department. 

Kind regards, 

on behalf of

Dr. Nicholas C. Manoukis 

Academic Editor

PLOS ONE